# Phenolic Compounds Content Evaluation of Lettuce Grown under Short-Term Preharvest Daytime or Nighttime Supplemental LEDs

**DOI:** 10.3390/plants11091123

**Published:** 2022-04-21

**Authors:** Aušra Brazaitytė, Viktorija Vaštakaitė-Kairienė, Rūta Sutulienė, Neringa Rasiukevičiūtė, Akvilė Viršilė, Jurga Miliauskienė, Kristina Laužikė, Alma Valiuškaitė, Lina Dėnė, Simona Chrapačienė, Asta Kupčinskienė, Giedrė Samuolienė

**Affiliations:** 1Lithuanian Research Centre for Agriculture and Forestry, Institute of Horticulture, Kaunas Str. 30, LT-54333 Babtai, Lithuania; viktorija.vastakaite-kairiene@lammc.lt (V.V.-K.); ruta.sutuliene@lammc.lt (R.S.); neringa.rasiukeviciute@lammc.lt (N.R.); akvile.virsile@lammc.lt (A.V.); jurga.miliauskiene@lammc.lt (J.M.); kristina.lauzike@lammc.lt (K.L.); alma.valiuskaite@lammc.lt (A.V.); lina.dene@lammc.lt (L.D.); simona.chrapaciene@lammc.lt (S.C.); asta.kupcinskiene@lammc.lt (A.K.); giedre.samuoliene@lammc.lt (G.S.); 2Department of Agroecosystems and Soil Sciences, Agriculture Academy, Vytautas Magnus University, Donelaicio Str. 58, 44248 Kaunas, Lithuania

**Keywords:** *Lactuca sativa* L., light-emitting diodes, phenolic acids, flavonoids

## Abstract

The study aimed to determine the changes in phenolic compounds content in lettuce (*Lactuca sativa* L. cv. Little Gem) depending on the preharvest short-term daytime or nighttime supplemental light-emitting diodes (LEDs) to high-pressure sodium lamps (HPS) lighting in a greenhouse during autumn and spring cultivation. Plants were grown in a greenhouse under HPS supplemented with 400 nm, 455 nm, 530 nm, 455 + 530 nm or 660 nm LEDs light for 4 h five days before harvest. Two experiments (EXP) were performed: EXP1—HPS, and LEDs treatment during daytime 6 PM–10 PM, and EXP2—LEDs treatment at nighttime during 10 AM–2 PM. LEDs’ photosynthetic photon flux density (PPFD) was 50 and HPS—90 ± 10 µmol m^−2^ s^−1^. The most pronounced positive effect on total phenolic compounds revealed supplemental 400 and 455 + 530 nm LEDs lighting, except its application during the daytime at spring cultivation, when all supplemental LEDs light had no impact on phenolics content variation. Supplemental 400 nm LEDs applied in the daytime increased chlorogenic acid during spring and chicoric acid during autumn cultivation. 400 nm LEDs used in nighttime enhanced chlorogenic acid accumulation and rutin during autumn. Chicoric and chlorogenic acid significantly increased under supplemental 455 + 530 nm LEDs applied at daytime in autumn and used at nighttime—in spring. Supplemental LEDs application in the nighttime resulted in higher phenolic compounds content during spring cultivation and the daytime during autumn cultivation.

## 1. Introduction

Nowadays, a growing interest in green eating is observed, affecting the increasing consumption of vegetables, including leafy vegetables. Therefore, not only their yield but also their nutritional quality becomes essential. Leafy vegetables are rich in bioactive secondary metabolites, which have health-beneficial properties for humans. Secondary metabolites participate in protecting plants against abiotic and biotic stresses and are essential for human nutrition, promoting the colour, taste, or aroma of plant products [1,2,3]. One of the leading secondary metabolites in plants are phenolic compounds. Such compounds, having antioxidant and anti-inflammatory properties, can help against the development of cancers, cardiovascular and neurodegenerative diseases, diabetes, obesity, etc. [3,4,5].

Because many leafy vegetables are grown in controlled environment agriculture (CEA), the synthesis and accumulation of secondary metabolites depend on the regulation of microclimate, remarkably light [1,3]. Plants react to light through multiple photoreceptors, which respond to a broad light spectrum, from ultraviolet B (UV-B) to far-red wavelengths and stimulate the biochemical pathways of such metabolites by regulating the expression of specific genes [2,3,6]. Moreover, in some cases, light could act as eustress (positive stress), stimulating the production of various phytochemicals, thus improving the nutritional value of leafy vegetables [1,3,7]. Nowadays, the application of the ecologically friendly technology of light-emitting diode (LED) lighting in CEA with its capability to select light wavelengths, change intensity, and reduce energy costs has many advantages compared to other conventional light sources. The ability to tailor the spectral composition according to plant or photoreceptor characteristics can affect leafy vegetables’ primary and secondary metabolic responses [3,8,9,10,11,12,13]. Moreover, monochromatic LEDs or their combination can be used to supplement the spectrum of high-pressure sodium (HPS) or fluorescent (FL) lamps, which are still widely applied in CEA, throughout the cultivation of vegetables or as short-term pre-harvest exposure [7,14,15,16,17,18,19,20]. Few studies concerning LEDs’ short-term pre-harvest exposure differ in the used spectrum, intensity, exposure time, and duration, and their effects on various phytochemical changes in leafy vegetables [7,15,16,17,18,20]. For example, short-term five days UV-A LEDs exposure increased antioxidant phenolic compounds in kale [20]. Short-term five days blue LED pre-harvest treatment significantly increased some carotenoids and glucosinolates in sprouting broccoli microgreens [21], and ten days exposure resulted in higher vitamin C, soluble protein, free amino acids, and chlorophyll in Chinese kale at harvest [22]. Continuous 48 h red-blue light-emitting diodes illumination depending on their ratio or intensity decreased nitrate and increased soluble sugars and vitamin C content [17,18]. Short-term pre-harvest red LED lighting exposure was shown as an efficient tool to reduce nitrate contents in various leafy vegetables [7,15] and produced baby leaf lettuce and *Brassicaceae* microgreens rich in total phenolics, tocopherols, sugars, and antioxidant capacity [16,23]. However, there is still a lack of information on how LEDs short-term pre-harvest exposure affects changes in the different phenolic compounds content in leafy vegetables. According to literature data, generally, different monochromatic light can affect the stimulation of secondary metabolites, especially phenolic compounds [2,3,24].

Furthermore, literature data showed that the growing season has also affected phytochemicals content in leafy vegetables [25]. Although vegetables are mainly grown in the CEA during the autumn-spring season, where supplemental lighting is used, limited data from the literature indicate the effect of seasonality. For example, after short-term red LED exposure, total phenolics content and DPPH free radical scavenging capacity within baby lettuces significantly increased during “dark” months, in November and January, but showed a different response in March [16]. Bioactive compounds increase in green baby leaf lettuces cultivar was observed in November and in red leaf cultivar in January under supplemental blue and green LEDs to HPS lighting [14]. The effect of seasonality on photosynthetic indices, growth, and phenolic compounds was determined in lamb’s lettuce under various LEDs light combinations [26,27].

Lettuce (*Lactuca sativa* L.) is one of the highly valuable leafy vegetables cultivated in CEA, mainly for its fresh leaves. Lettuce varies in colors, sizes, shapes, and is mostly used in salad mixes. It is low in calories, fat, Na, a good source of fiber, minerals, various vitamins and bioactive compounds such as folate, vitamin E, vitamin C, β-carotene, and phenolic compounds. Phenolic acids, especially caffeic acid, chlorogenic acid, and their derivatives, and flavonoids such as quercetin and kaempferol derivatives, anthocyanins, and flavone luteolin are reported as the main phenolic compounds in lettuce [28]. Furthermore, literature data showed that phenolic compounds content in lettuce could be enhanced by manipulating various agricultural practices, including light through the application of LEDs [12,16,19,28]. For example, anthocyanin, flavonoids, and chlorogenic acids in red leaf lettuce significantly increased under a higher percentage of blue light [12]. In the same type of lettuce, the predawn application of blue light showed the highest content of phenolic acids and flavonoids in comparison with green leaf lettuce [19]. Supplemental red-LEDs before harvesting resulted in an increase in the total phenolics of baby leaf lettuce [16]. Therefore, we hypothesised that even short-term exposure of LEDs light as supplemental to HPS lighting would positively affect the content of phenolic compounds in lettuce depending on the time of day or season. Thus, our study aimed to determine the changes of phenolic compounds content in lettuce depending on the short-term daytime or nighttime preharvest supplemental LEDs to HPS lighting in a greenhouse during autumn and spring cultivation.

## 2. Results

### 2.1. Effect of Short-Term Daytime Supplemental LEDs to HPS Lighting on Phenolic Compounds Content in Lettuce Cultivated in A Greenhouse during Different Seasons

The results that short-term daytime preharvest supplemental LEDs to HPS lighting in a greenhouse during spring cultivation did not affect total content of phenolic compounds in lettuce (Table 1). There was a trend, that different lighting had different effects on the content of individual phenolic compounds in lettuce. Although daytime supplemental LEDs light during spring cultivation of lettuce reduced or did not affect the content of many phenolic compounds, chlorogenic acid was significantly increased under supplemental 400 nm LEDs, and rosmarinic acid—under 530 nm LEDs. Also, it was determined the positive effect of supplemental 455 nm LEDs on chlorogenic and rosmarinic acid content in lettuce. Meanwhile, compared to HPS lighting, supplemental 660 nm LEDs resulted in the lowest gallic acid and apigenin, myricetin, and rutin content, 530 nm—gallic acid and apigenin, epicatechin, quercetin, rutin, 455 + 530 nm—protocatechuic and rosmarinic acids, apigenin and myricetin, 455 nm—protocatechuic, kaempferol, 400 nm—rutin. All supplemental LEDs decreased caffeic and p-coumaric acids content.

During cultivation in autumn, the more evident effect of different lighting on the phenolic compounds in lettuce was noticed. Total phenol compounds content as well as phenolic acids such as caffeic, chicoric, chlorogenic, rosmarinic, and flavonol quercetin content were higher under supplemental 455 + 530 nm LEDs than under HPS lighting alone. Supplemental 660 nm LEDs significantly increased caffeic and o-coumaric acids, epicatechin, quercetin and rutin. Positive effects on the increase of chicoric and o-coumaric acids, epicatechin and quercetin, were determined under supplemental 400 nm and rosmarinic acid under 455 nm LEDs light. Although supplemental 530 nm LEDs light increased p-coumaric and rosmarinic acid content, the lowest content of caffeic, chlorogenic, protocatechuic acids and quercetin, and rutin was established. Different lighting had no significant effect on gallic acid, kaempferol, and myricetin content in lettuce.

The incidence of significant light and season interaction (LxS) indicates differential response to short-term daytime supplemental LEDs light treatments at spring and autumn examined with respect to phenolic compounds content (Table 1). However, the relative contribution of the main effects to the variance of phenolic compounds indicates that variation is introduced principally by season (S) (Appendix A) and much less by light treatment (L) (Appendix A). Season had no effect only on caffeic acid content. Higher levels of chicoric, chlorogenic and o-coumaric acids and epicatechin, quercetin, rutin and total phenolics were found during autumn cultivation (Appendix A). Meanwhile, the effect of light treatment (L) was more pronounced on caffeic acid and epicatechin, rutin content, which were higher under the 660 nm supplemental LEDs treatments and quercetin content under 400 nm in comparison with other light treatments (Appendix A).

### 2.2. Effect of Short-Term Nighttime Supplemental LEDs to HPS Lighting on Phenolic Compounds Content in Lettuce Cultivated in A Greenhouse during Different Seasons

According to data obtained from experiments, when short-term supplemental LEDs light were applied at nighttime, the most positive effect for total phenolic compounds increase was found under 455 + 530 nm LEDs during both cultivation periods and under 400 nm LEDs during autumn cultivation (Table 2). Generally, higher total phenolic compounds content in lettuce was determined under spring cultivation, contrary to what supplemental LEDs were applied in the daytime.

During spring cultivation, the supplemental 455 + 530 nm LEDs at nighttime increased the content of phenolic acids such as chicoric, chlorogenic, rosmarinic, and flavonol apigenin, which were the main part of the total phenolic compounds. Supplemental 455 nm LEDs light increased caffeic acid and 660 nm—kaempferol content. Different lighting had no significant effect on p-coumaric acid and rutin content. Gallic, o-coumaric, protocatechuic acids, and epicatechin decreased under all supplemental LEDs light. Supplemental 530 nm LEDs light resulted in the significantly lowest content of chicoric, chlorogenic, o-coumaric, and protocatechuic acids, 455 + 530 nm—caffeic acid, 455 nm—gallic acid and quercetin, 400 nm—epicatechin compared to HPS lighting.

Although nighttime supplemental LEDs light during autumn cultivation of lettuce did not affect many phenolic compounds content, chlorogenic, gallic, and o-coumaric acids, myricetin, and rutin increased under supplemental 400 nm LEDs as well as rutin under 455 + 530 nm LEDs. The positive effect of supplemental 530 nm LEDs was determined on the increase of caffeic acid. Meanwhile, supplemental 455 nm LEDs light significantly decreased chlorogenic acid and rutin content.

The same as at daytime application, significant light and season interaction (L × S) were determined on phenolic compounds content (Table 2). Season had no effect on o-coumaric acid and myricetin content. Higher content of total phenolics and practically all individual phenolic compounds, except o-coumaric acid and rutin were found during spring cultivation (Appendix A). Meanwhile, the effect of light treatment (L) was more obvious on chlorogenic, o-coumaric acids, epicatechin and total phenolics content (Appendix A).

## 3. Discussion

According to various literature sources, phenolic compounds show plasticity in response to light quality, quantity, and duration, allowing plants to adapt to their changes and act as sunscreen, antioxidants, or both. Many studies related to the light quality concern UV-A and blue light as having the most effective impact on phenylpropanoid metabolism than the other light spectrum [6,11,14,19,22,24,29,30]. It was determined that such light stimulates the genes expression belonging to the phenylpropanoid pathway, which is involved in the biosynthesis of phenolic acids and flavonoids mediated by cryptochromes [11,22,24,29,31,32]. In the present study, we used supplemental LEDs of 455 nm, and 400 nm wavelengths, which also could be attributed to the UV-A light spectrum [20,24]. Our data showed that monochromatic supplemental 400 nm LEDs light positively affected the accumulation of total phenolic compounds in lettuce during autumn cultivation but not spring. However, its effect on individual phenolic compounds differed. The significantly higher content of chlorogenic acid and rutin, which were the main part of total phenolic compounds, were determined when such light was applied at nighttime.

Meanwhile, 400 nm LEDs light application at daytime positively affected chichoric acid, epicatechin, and quercetin. O-coumaric acid content increased in both cases. Other authors also reported that a shorter blue or UV-A wavelength enhanced the accumulation of the phenolic compounds in various plants. For example, Lee and coauthors [20] stated that short-term 385 nm UV-A exposure resulted in a significant increase of total phenols, caffeic acid, and kaempferol, but not ferulic acid. Treatments with specific white LEDs light contained a shorter blue wavelength enhanced the accumulation of the individual compounds in butterhead and romaine lettuce cultivars compared to longer ones [30] as well as total phenolics and flavonoids in pak choi [31]. Taulavuori and coauthors [32] showed that violet (420 nm) containing blue (440 nm) light was slightly more effective in the stimulation of flavonoid synthesis in arugula than only blue (450 nm) light. It is known that some flavonoids mostly absorb at 400–430 nm wavelengths light, so it was presumed that shorter blue wavelengths with higher energy could efficiently promote phenolic acid and flavonoid accumulation [30,33]. 

Meanwhile, supplemental 455 nm LED light applied at nighttime significantly decreased and used at daytime only slightly increased total phenolic compounds content compared to HPS lighting. Such light effect on individual phenolic compounds depended on application time and season. For example, supplemental 455 nm LEDs light at daytime enhanced caffeic acid accumulation in lettuce during autumn cultivation and applied at nighttime during spring cultivation. Meanwhile, chlorogenic acid content increased during spring cultivation when 455 nm LEDs light was used in the daytime and positively affected rosmarinic acid content in all cases. Other authors noticed the positive impact of longer blue wavelengths (450–470 nm) on the accumulation of phenolic compounds in leafy vegetables such as different lettuces varieties, pak choi, Chinese kale, basil, etc. depending on exposure duration till harvest, photoperiod during the daytime, intensity [6,11,14,19,33,34]. Various authors showed that the transcriptional levels of flavonoid biosynthetic genes are strongly affected by the time duration, amount, and blue or UV-A wavelength of light [24,31,33]. According to our results, it can be assumed that 400 nm light stimulates the genes expression belonging to the phenylpropanoid pathway more in comparison to 455 nm. On other hand, our results would suggest that the photoperiod of 455 nm LEDs exposure could be longer and/or higher light intensity for stimulation such genes expression and enhancing phenolic compounds accumulation in lettuce, but further studies are needed. 

It is known that green light, the same as blue and UV, is also absorbed by the cryptochromes, although the specific photoreceptor of such light remains to be identified in higher plants. The green light can be more efficiently absorbed by the outer leaves of the canopy and stimulate photosynthesis at lower leaf levels [6,11,24]. However, green LEDs are not extensively applied in commercial plant cultivation due to the inefficiency in converting electricity into photons [8,29,35]. Therefore, there is comparatively little literature on its impacts on plants quality, including phenolic compounds. Few accessible reports showed that green light used as monochromatic or as part of a broader light combination frequently had no effect or reduced accumulation of phenolic compounds and can reverse the positive impact of monochromatic blue light [6,11,14,24,36,37,38,39]. The green light had positive results in only a few cases. For example, it enhanced total phenolic and total flavonoids production of *Prunella vulgaris* callus cultures [40]. The significant increase of total phenols in green baby leaf lettuce cultivated in January was found under supplemental 535 nm LEDs and HPS lighting [14]. The present study showed no/or negative effect of green light on total phenolic compounds content in lettuce depending on cultivation season. However, there is a lack of literature data about its impacts on individual phenolic compounds. Our data revealed the positive effect of supplemental green light applied in the daytime on rosmarinic acid during both cultivation seasons. It also enhanced the accumulation of quercetin during spring and p-coumaric acid during autumn cultivation. Meanwhile, the green light at nighttime showed more negative effects than daytime application, except caffeic acid, which significantly increased during autumn application.

However, this study revealed the different effects of green light applied with blue in equal proportions (455 + 530 nm). Such combined lighting used in the daytime during autumn cultivation enhanced the accumulation of total phenolic compounds like chicoric, chlorogenic, and rosmarinic acids, which were mainly in total content. Meanwhile, application at nighttime positively affected total phenolic compounds and the above-mentioned phenolic acids during both cultivation seasons and apigenin during spring and rutin during autumn cultivation. It could show that supplemental blue-green light in a darker period of the day or season was more efficient on phenolic compounds accumulation. Meanwhile, such lighting revealed a more evident positive effect than only blue supplemental LEDs light. According to literature, shorter green light wavelengths less than 530 nm are perceived as part of the cryptochrome and phototropin blue light response. However, longer green light wavelengths about 570 nm are sufficient to antagonise blue light cryptochrome activation [41,42,43]. In our experiments, we used shorter 530 nm green light, which applied with blue light could stimulate the genes expression belonging to the phenylpropanoid pathway similar to UV-A and blue light and enhanced biosynthesis of phenolic acids and flavonoids in lettuce. We did not find other data concerning the effect of blue/green light on polyphenolic content and composition in leafy vegetables. However, Zheng and coauthors [44] reported that, when dichromatic blue-green light in equal proportion was applied for 4 h in the nighttime, the expression level of several key structural genes among the flavonoid biosynthesis pathway in tea plants decreased compared to the application of monochromatic blue or green light. Therefore, the accumulation of anthocyanin and catechins in such plants under blue-green light was lower than under blue light alone [44].

On the other hand, it was reported that plants interpret decreased blue: green ratios as a shade response, which could act as abiotic stress [42]. It is known that phenolic compounds are produced in plants to overcome potential stressful conditions [45,46]. Therefore, supplemental blue-green light to HPS in which spectrum dominated red-orange-yellow, could change this ratio into a decrease and may act as eustress leading to an increase in phenolic compounds content in lettuce. But further research is required to confirm this assumption. 

The present study showed that supplemental 660 nm LEDs light decreased or had no significant effect on total phenolic compounds content compared to HPS lighting. However, some phenolic acid such as caffeic, o-coumaric, and flavonoids epicatechin and rutin significantly increased when the supplemental red light was applied in the daytime during autumn cultivation. According to most studies, red light does not promote phenylpropanoid stimulation of polyphenol biosynthesis, but some cases showed the positive effect of such compounds [2,6,16,23,24]. However, red light’s impact on the phenolic accumulation in plants depended on the species and variety, leaf age, duration of exposure, etc. For example, sole red light caused an increase of phenolic compounds in the young red and green Perilla leaves, but not in the mature leaves [47]. The 3-day pre-harvest red (638 nm, 300 μmol m^−2^ s^−1^) light exposure has a more evident effect on the level of total phenolics in baby leaf lettuce grown in greenhouse conditions in winter [16] and mustard [48]. The higher percentage of red light in red-blue lighting resulted in higher gallic acid and quercetin content in the green basil cultivar, but not in red [49]. The supplemental red light was most effective in enhancing the accumulation of chlorogenic, caffeic, and chicoric acids, rutin, kaempferol, and luteolin in red leaf lettuce [50]. In all of these studies reviewed, red light exposure was longer and more intense during the day. Therefore, it can be assumed that the photoperiod of 660 nm LEDs exposure could be longer and/or higher light intensity for more evidence effect on the increase of polyphenol content.

Generally, in our studies, it was observed that although the variation of the total content of phenolic compounds depending on the lighting and growing season was not large, the content of individual phenolics differed several times. Other authors also determined such trend in lettuce plants [50,51]. It was suggested that it is probably related with enhanced activity of the phenylpropanoid pathway, resulting in an increase of intermediate and final products of this branched metabolic pathway. For example, chlorogenic acid characterizes a typical compound synthesized within the phenylpropanoid pathway and relates to p-coumaric acid as an intermediate product [51].

Few studies have described the seasonal effect on phenolic compounds and their individual composition in leafy vegetables, suggesting that variation in phenolic compounds is more dependent on growing conditions and cultivar [14,16,27,52,53,54]. For example, it was observed that differences among five lettuce cultivars appear to have a more significant impact on phenolic compounds than environmental variation during the growing season [52]. The best light combination for increasing phenolic compounds content in Lamb’s lettuce grown in the greenhouse was 70R/30B in autumn and 50R/50B in winter cultivation [27]. Samuoliene and coauthors [14,16] showed that total phenols content in various baby leaf lettuces mainly increased in the darker months—November and January—when the supplemental blue, green or red light was applied together with HPS lighting. Marin and coauthors [53] observed that the increase in temperature and radiation from February to May promoted the increase in the content of phenolic acids and flavonoids and showed the seasonal variations of individual phenolic compounds. Meanwhile, Lee and coauthors [54] reported that analysed phenolic acids in red Chinese cabbages increased in autumn and flavonols in spring cultivation. The present study showed a significant seasonality effect, but it depends on supplemental LEDs light application time. The application of LEDs during the daytime resulted in higher total phenolic compounds content during autumn cultivation. However, individual phenolic compounds such as gallic, protocatechuic, p-coumaric, rosmarinic acids, myricetin, apigenin, and kaempferol content were higher during spring cultivation. Meanwhile, supplemental LEDs application in the nighttime increased analysed phenolic compounds content during spring cultivation, except rutin. Generally, our results confirm other authors observations that increased radiation in spring increasing phenolics content maybe due to higher stimulation the genes expression of phenylpropanoid pathway [33,53,54]. That suggests different lighting strategies for increasing phenolic compounds content in lettuces during different growing seasons, but further studies are need for better understanding of the regulation phenolics compounds synthesis during different growing seasons 

## 4. Materials and Methods

### 4.1. Growth Conditions

Pot experiments were performed at the greenhouse of the Institute of Horticulture, Lithuanian Research Centre of Agricultural and Forestry (lat. 55° N), during the autumn and spring periods. Seeds of lettuce (*Lactuca sativa* L. cv. Little Gem; CN Seeds, UK) were sown in cell trays (70 mL cell volume, three seeds per tray, 54 trays in one plastic vessel) containing peat substrate (Terraerden, Rucava, Latvia) with NPK (100–160; 110–180; 120–200 mg L^−1^) and microelements Mn, Cu, Mo, B, Zn and Fe (pH H_2_O 5.5–6.5; electrical conductivity (EC) ms cm^−1^ < 1.10). Seedlings with one true leaf (BBCH 11–12) were transplanted into pots 500 mL, filled with the same peat substrate. Plants were watered when needed, maintaining a similar substrate humidity. From the 10th day after transplanting (BBCH 14–15), plants were watered with 50 mL of nutrient solution for plant (10 mL NPK 3-1-3 (Terra grow, Plagron, Netherlands) in 5 L water) two times a week. The day/night temperatures of 23 ± 2/16 ± 2 °C, and 16-h photoperiod and relative air humidity of 70 ± 10% were maintained. Plants were grown under daylight with supplementary lighting provided by standard high-pressure sodium lamps (HPS) (SON-T Agro, 400 W, Philips, Eindhoven, The Netherlands) 16-h photoperiod. The generated photosynthetic photon flux density (PPFD) of HPS lamps at plant level was about 90 ± 10 μmol m^−2^ s^−1^. During the experiments, the weekly-average solar radiation inside the greenhouse ranged from 20 to 80 μmol m^−2^ s^−1^ in November–December and from 150 to 230 μmol m^−2^ s^−1^ in March–April. At thirty days after germination, lighting experiments began (see Section 4.2). At the end of the experiments, plants were harvested just above the substrate level. Samples of randomly selected twelve fully developed lettuce plants per treatment were used for phytochemical analysis.

### 4.2. Lighting Treatments

At the lettuce pre-harvest stage of 5 days, the HPS lamps were supplemented by LEDs lamps (4h photoperiod) (Figure 1 and Figure 2). Different LEDs lamps (Vegetal Grow Development, France) contained diodes with the peak wavelength of UV-A (400 nm), blue (455 nm), green (530 nm), blue + green (455 + 530 nm) and red (660 nm). Two lighting experiments (EXP1 and EXP2) with replication were carried out during the autumn and spring periods. EXP1 − HPS + LEDs lamps lighting was from 06 till 10 AM where HPS lamps generated 90 ± 10 μmol m^−2^ s^−1^ and LEDs generated 50 μmol m^−2^ s^−1^ (Figure 1). Later, only HPS lighting was from 10 AM till 10 PM. EXP2—HPS lighting was from 06 AM till 10 PM, and LEDs lamps lighting was from 10 PM till 02 AM (Figure 2). A photometer RF-100 with head G.PAR-100 was used to measure PPFD (Sonopan, Bialystok, Poland).

### 4.3. Determination of Individual Phenolic Compounds

The fresh plant material was immediately frozen in liquid N_2_ and lyophilised for in-dividual phenolic compound analysis. It was calculated that the average content of dry matter in lettuce was about 4.5% during both growing seasons. Individual phenolic compounds were analysed by high-performance liquid chromatography (HPLC) on a NUCLEODUR Sphinx RP column (5 µm particle size, 150 × 4.6 mm) (Macherey-Nagel GmbH & Co KG, Düren Germany). For phenolic compounds extraction, 100 mg of lyophilised plant material was grounded with 80% ice-cold methanol (Sigma-Aldrich, St. Louis, MO, USA) and transferred to a 15 mL polypropylene conical centrifuge tube (Labbox Labware S.L., Barcelona, Spain). The extract was incubated at 4 °C for 24 h. Samples were centrifuged (Hermle Z 300 K, Hermle Labortechnik, Wehingen, Germany) at a relative centrifugal force of 4000 rpm min^−1^ for 10 min at room temperature. The supernatant was filtered through a 70 mm qualitative filter paper (Frisenette ApS, Knebel, Denmark). Before the HPLC analyses, samples were filtrated through a 13 mm and 0.22 µm nylon syringe filter (BGB Analytik AG, Böckten, Switzerland). The HPLC 10A system (Shimadzu, Kyoto, Japan) equipped with a diode array (SPD-M 10A VP) detector was used for analysis. Peaks were detected at 280 nm. The mobile phase consisted of A (100% acetonitrile, (Sigma-Aldrich, St. Louis, MO, USA) and B (1% acetic acid, Supelco, Bellefonte, PA, USA). Binary gradient: 0 min; 95% B, 25 min; 70% B, 25–30 min; 5% B, 30–35; 5% B; 35–37 min; 95% B, and 37–40 min; 95% B, flow rate 1 mL min^−1^. The results are expressed as an average of analytical measurements of three biological samples from homogenized plant material in mg g^−1^ in the dry mass of plants. The contents of rutin (rutin trihydrate, Supelco), myricetin, chicoric acid, ferulic acid (*trans*-Ferulic acid), rosmarinic acid, quercetin and protocatechuic acid (all purchased from Merck KGaA, Darmstadt, Germany), caffeic acid, p-coumaric acid (*trans*-p-Coumaric acid), o-coumaric acid (*trans*-2-Hydroxycinnamic acid), m-coumaric acid (*trans*-3-Hydroxycinnamic acid), epicatechin, chlorogenic acid, kaempferol, gallic acid (gallic acid monohydrate) (all purchased from Sigma-Aldric), and apigenin (LGC Standards Ltd., LGC, Teddington, UK) are expressed as mg g^−1^ in the dry matter of plants.

### 4.4. Statistical Analysis

Statistical analysis was performed using Microsoft Excel 2016 and Addinsoft XLSTAT 2022 XLSTAT statistical and data analysis (Long Island, NY, USA). Two-way analysis of variance (ANOVA) followed by Tukey’s honestly significant difference test (*p* < 0.05) for multiple comparisons was used to evaluate differences between means (*n* = 3) of measurements. 

## 5. Conclusions

This study demonstrated that even short-term supplemental LEDs preharvest lighting affected the accumulation of total and individual phenolic compounds in lettuces. However, our results suggest different LEDs application strategies for increasing their content during different growing seasons and times of the day. The most pronounced positive effect on total phenolic compounds revealed supplemental 400 and 455 + 530 nm LEDs lighting, except its application during the daytime during spring cultivation, when all supplemental LEDs light had no impact on such compound. Supplemental 400 nm LEDS applied in the daytime increased primary phenolic compounds such as chlorogenic acid during spring and chicoric acid during autumn cultivation. Meanwhile, 400 nm LEDs used in nighttime enhanced chlorogenic acid accumulation and rutin during autumn but were not effective during spring cultivation. Chicoric and chlorogenic acid significantly increased under supplemental 455 + 530 nm LEDs applied at daytime in autumn and used at nighttime—in spring. Supplemental LEDs application in the nighttime resulted in higher analysed phenolic compounds content during spring cultivation, except rutin. When applied in the daytime, higher total phenolic compounds, chicoric and chlorogenic acid content were determined during autumn cultivation. A review of our and the literature data suggests that further research is required to clarify the impact of more prolonged and intense supplemental LEDs to HPS lighting exposure on various phenolic compounds accumulation and biosynthetic pathways in lettuce.

## Figures and Tables

**Figure 1 plants-11-01123-f001:**
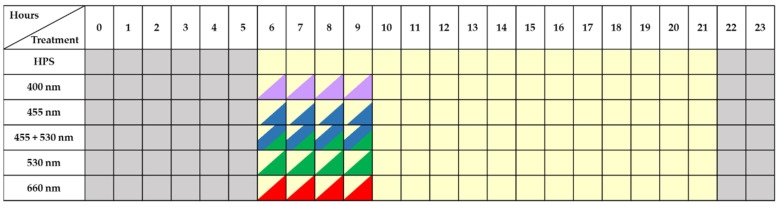
The lighting scheme of 1st experiment (EXP1).

**Figure 2 plants-11-01123-f002:**
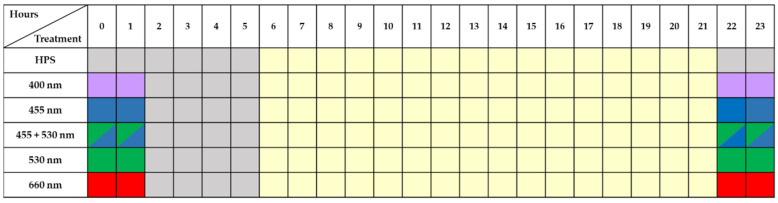
The lighting scheme of the 2nd experiment (EXP2).

**Table 1 plants-11-01123-t001:** Effect of short-term daytime supplemental LEDs to HPS lighting on phenolic compounds content in lettuce cultivated in a greenhouse during different seasons.

Phenolic Compounds	Lighting												Source of Variance		
	HPS		400 nm		455 nm		455 + 530 nm		530 nm		660 nm		L	S	L ×S
	Spring	Autumn	Spring	Autumn	Spring	Autumn	Spring	Autumn	Spring	Autumn	Spring	Autumn			
Caffeic a.	0.068 b	0.015 gh	0.033 ef	0.040 def	0.047 de	0.05 cd	0.029 efg	0.066 bc	0.025 fg	0.007 h	0.042 def	0.151 a	*		*
Chicoric a.	2.57 cd	3.89 bc	2.43 d	5.07 ab	2.24 d	3.83 bc	2.57 cd	5.51 a	2.26 d	3.87 bc	1.97 d	2.73 cd		*	*
Chlorogenic a.	0.81 de	1.04 bc	1.07 bc	0.98 cd	0.98 cd	1.16 bc	0.74 e	1.46 a	0.77 e	0.72 e	0.75 e	1.22 b		*	*
Gallic a.	0.049 a	0.029 bc	0.034 abc	0.020 bc	0.037 ab	0.019 bc	0.031 abc	0.022 bc	0.028 bc	0.018 c	0.026 bc	0.020 bc		*	*
o-coumaric a.	0.038 c	0.141 b	0.034 c	0.230 a	0.042 c	0.057 c	0.056 c	0.038 c	0.044 c	0.043 c	0.039 c	0.227 a		*	*
p-coumaric a	0.082 a	0.008 f	0.052 b	0.016 ef	0.039 c	0.016 ef	0.030 cd	0.010 ef	0.030 cd	0.033 c	0.021 de	0.016 ef		*	*
Protocatechuic a.	0.177 a	0.071 c	0.138 b	0.023 d	0.133 b	0.022 d	0.073 c	0.023 d	0.160 ab	0.020 d	0.142 ab	0.093 c		*	*
Rosmarinic a.	0.641 bc	0.049 f	0.596 c	0.076 f	0.732 a	0.520 cd	0.563 c	0.424 d	0.797 a	0.235 e	0.601 c	0.033 f		*	*
Apigenin	0.844 a	0.333 e	0.637 bc	0.082 f	0.725 ab	0.358 e	0.458 de	0.044 f	0.715 b	0.381 e	0.521 cd	0.089 f		*	*
Epicatechin	0.165 cd	0.280 b	0.080 de	0.432 a	0.069 de	0.137 cde	0.080 de	0.199 bc	0.047 e	0.187 bc	0.123 cde	0.515 a	*	*	*
Kaempferol	0.034 ab	0.020 cd	0.038 a	0.013 d	0.015 cd	0.026 bc	0.026 bc	0.020 cd	0.035 ab	0.024 bcd	0.021 cd	0.022 cd		*	*
Myricetin	0.118 a	0.051 c	0.089 b	0.035 c	0.107 ab	0.033 c	0.053 c	0.038 c	0.099 ab	0.037 c	0.052 c	0.040 c		*	*
Quercetin	0.043 d	0.022 de	0.036 de	0.180 a	0.026 de	0.015 e	0.036 de	0.179 a	0.071 c	0.014 e	0.028 de	0.139 b	*	*	*
Rutin	0.026 b	0.042 b	0.013 c	0.061 b	0.026 b	0.046 b	0.027 b	0.049 b	0.016 c	0.019 c	0.012 c	0.361 a	*	*	*
Total	5.67 cde	5.99 cd	5.28 cde	7.25 ab	5.21 cde	6.29 bc	4.78 de	8.08 a	5.10 cde	5.61 cde	4.34 e	5.65 cde		*	*

L—lighting; S—seasons; a.—acid. Individual phenolic compound content is presented as mg g^−1^ in dry plant matter. Means with different letters (based on heatmap values with the same letters on a separate line are marked with the same color) are significantly different at the *p* < 0.05 level by Tukey’s honestly significant difference test (*).

**Table 2 plants-11-01123-t002:** Effect of short-term nighttime supplemental LEDs to HPS lighting on phenolic compounds content in lettuce cultivated in a greenhouse during different seasons.

Phenolic Compounds	Lighting												Source of Variance		
	HPS		400 nm		455 nm		455 + 530 nm		530 nm		660 nm		L	S	L ×S
	Spring	Autumn	Spring	Autumn	Spring	Autumn	Spring	Autumn	Spring	Autumn	Spring	Autumn			
Caffeic a.	0.222 b	0.036 ef	0.217 b	0.043 e	0.269 a	0.018 ef	0.116 d	0.039 e	0.181 c	0.098 d	0.180 c	0.009 f		*	*
Chicoric a.	4.47 b	0.28 e	2.40 d	0.40 e	3.65 c	0.08 e	5.42 a	0.37 e	1.78 d	0.29 e	1.89 d	0.13 e		*	*
Chlorogenic a.	2.59 b	1.11 de	1.39 d	2.15 c	2.04 c	0.33 f	5.08 a	1.44 d	1.25 d	1.12 de	2.01 c	0.80 e	*	*	*
Gallic a.	0.059 a	0.012 f	0.030 cd	0.036 bc	0.026 cde	0.016 def	0.049 ab	0.013 ef	0.044 b	0.009 f	0.027 cd	0.017 def		*	*
o-coumaric a.	0.303 ab	0.190 bc	0.043 c	0.382 a	0.070 bc	0.035 c	0.087 bc	0.047 c	0.035 c	0.115 bc	0.036 c	0.067 bc	*		*
p-coumaric a	0.021 ab	0.005 ab	0.021 ab	0.027 ab	0.025 ab	0.005 ab	0.052 a	0.003 b	0.024 ab	0.003 b	0.038 ab	0.004 ab		*	*
Protocatechuic a.	0.210 a	0.012 d	0.122 bc	0.022 d	0.101 c	0.013 d	0.148 b	0.019 d	0.098 c	0.022 d	0.156 b	0.029 d		*	*
Rosmarinic a.	0.098 de	0.006 e	0.723 c	0.015 e	1.153 b	0.007 e	2.318 a	0.005 e	0.453 cd	0.006 e	0.771 c	0.006 e		*	*
Apigenin	0.390 b	0.010 e	0.428 b	0.051 e	0.255 cd	0.044 e	0.597 a	0.010 e	0.298 c	0.009 e	0.196 d	0.061 e		*	*
Epicatechin	0.493 a	0.095 bcd	0.088 cd	0.056 d	0.132 bc	0.050 d	0.146 b	0.067 d	0.480 a	0.078 d	0.143 b	0.050 d	*	*	*
Kaempferol	0.012 bc	0.002 de	0.012 bcd	0.005 cde	0.014 b	0.003 cde	0.016 b	0.004 cde	0.012 bc	0.001 e	0.031 a	0.002 e		*	*
Myricetin	0.031 abcd	0.027 bcd	0.027 bcd	0.052 a	0.029 bcd	0.017 cd	0.036 abc	0.014 d	0.045 ab	0.011 d	0.020 cd	0.015 d			*
Quercetin	0.025 ab	0.005 d	0.018 c	0.005 d	0.017 c	0.005 d	0.019 bc	0.006 d	0.029 a	0.005 d	0.023 abc	0.005 d		*	*
Rutin	0.179 f	2.809 c	0.041 f	5.101 a	0.062 f	1.299 e	0.091 f	3.390 b	0.053 f	2.478 cd	0.027 f	2.187 d		*	*
Total	9.11 b	4.59 ef	5.56 d	8.35 bc	7.85 c	1.92 g	14.17 a	5.43 d	4.78 de	4.24 ef	5.55 d	3.38 f	*	*	*

L—lighting; S—seasons; a.—acid. Individual phenolic compound content is presented as mg g^−1^ in dry plant matter. Means with different letters (based on heatmapvalues with the same letters on a separate line are marked with the same color) are significantly different at the *p* < 0.05 level by Tukey’s honestly significant difference test (*).

## Data Availability

Not applicable.

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
