# Peer review of "Phenolic Compounds Content Evaluation of Lettuce Grown under Short-Term Preharvest Daytime or Nighttime Supplemental LEDs"

_plants, 2022, doi:10.3390/plants11091123_

Round 1

Reviewer 1 Report

Comments

The manuscript entitled “Polyphenols Content Evaluation of Lettuce Grown under Short- 2 term Preharvest Daytime or Nightime Supplemental LEDs”  is an important research work for the micro green research field using LEDs. There are some issues that required immediate attention prior to its acceptance.

Abstract: Avoid using unnecessary marking.  Example: Is it 400 minus or something else?

400-, 455-, 530-, 455+530-

Line 24: write the photoperiod value correctly

“treatment at night during 22-02 h.”

Line 31: Check

“used at nighttim – in spring.”

Introduction

Line 38: Leafy.. what?????

“including leafy.”

Line 83-85: rewrite the sentence.

“In addition, the effect of seasonality on bioactive compounds in baby leaf lettuces was determined under supplemental blue, and green  LEDs to HPS lighting [14] and on photosynthesis indices, growth, and polyphenols in  lamb's lettuce under various LEDs light combinations [26, 27].”

Write the advantages of using LEDs over conventional light sources.

Write more about “Lettuce (Lactuca sativa L.)”

Results

Line 100:  “assaying”

Discussion:

Except in some cases, seasonal variation was more distinct in the result. Discuss it in the text citing recent work.

The proper mechanism responsible for increasing the total phenolic compounds by LEDs or its supplemented with HPS is missing in the text. Discuss it.

Material methods:

From where the chemicals and plant seeds were obtained? Mention it

Reviewer 2 Report

The authors present interesting results on the accumulation of phenolic compounds under supplemental LED illumination. Two types of experiments were conducted when LED light was applied to plants short before harvest during the day or at night. In addition, the experiments were carried out in spring, when there was more natural sunlight, and in autumn, when there was less sunlight. During cultivation, the plants were lighted with HPS lamps. The work is very interesting and can be very useful in horticultural practice, but I have some comments.

In my opinion, the authors must not use the term polyphenols, but should use phenolic compounds or phenolics. Polyphenols are substances containing more than one aromatic ring with an OH group attached. The authors designate, inter alia, simple phenolic acids which cannot be included in this group. I therefore propose that the title and the text of the paper should be corrected.

I am not sure if phrase 'green eating' is corrected, for me is too colloquial.

I am not sure if addition 'internal' to quality is necessary.

line 40 better will be : health-beneficial properties for humans

line 40-41 - small-scale and low concetration - in compare to what? This sentence needs rewording.

line 63 - 66: do the authors mean the fifth day or five days - because it seems to me that the abbreviation is used incorrectly

line 67: a sentence should not begin with a number

line 85: in my opinion better will be photosynthetic indices

line 89: all nutrients should be healthy so for me source of nutrients is better

results: I have some doubts as to whether the assumptions of the analysis of variance were met and whether the authors applied this statistical test well. In my opinion, the assumption of normal distribution was not met in many cases, and I am also curious as to whether the variance was homogeneous. If the anova assumptions were met - the authors chose to show the results of a two-factor analysis of variance. For each parameter presented, the authors show that there is an interaction between two factors (light x season), but the results are discussed separately for autumn and spring. Neither in the results chapter nor in the discussion chapter do the authors refer to interaction - there is only a brief statement: a different reaction. Why this is so I have not found anywhere. I realise that this is a difficult question, but looking at how the results chapter is written I would suggest to the authors that they present their achievements a little differently. Please consider this solution: present the results for spring first (all types of experimentation) and then for autumn - this will make it easier to follow the results together with the text of the chapter. As homogeneous groups are combined and sometimes there are even 3 letters with mean, it is possible to introduce heat map elements, i.e. mark contents significantly higher than in HPS with one colour and contents lower than in HPS with another colour, leave no differences on white background - no colour - maybe this will make reading these very extensive tables easier.

In my opinion, the data with the percentage of dry matter in the tissues of the tested plants is very lacking, do the authors have this data and can they attach it to the paper?

line 104: Sentence too general and I cannot agree with it, needs rewording

In the results themselves I am sometimes surprised by the values, after adding LED radiation to HPS sometimes the authors obtain even a 10-fold difference in the content of a given compound - this is very high - but in the discussion there is no hypothesis as to why this may happen.

In my opinion, the conclusions section should be called summary

I have inserted a few comments in the text and highlighted in colour where there is a need for improvement.

In my opinion the work is interesting and after additions and corrections it will be suitable for publication.

Round 2

Reviewer 1 Report

The author of the manuscript has revised the research paper carefully and responded to all the queries appropriately. The manuscript can be accepted in the present form.